# Food Sources of Shortfall Nutrients among Latin Americans: Results from the Latin American Study of Health and Nutrition (ELANS)

**DOI:** 10.3390/ijerph18094967

**Published:** 2021-05-07

**Authors:** Ana Carolina Barco Leme, Regina Mara Fisberg, Aline Veroneze de Mello, Cristiane Hermes Sales, Gerson Ferrari, Jess Haines, Attilo Rigotti, Georgina Gómez, Irina Kovalskys, Lilia Yadira Cortés Sanabria, Marianella Herrera-Cuenca, Martha Cecília Yépez Garcia, Rossina G. Pareja, Mauro Fisberg

**Affiliations:** 1Center for Excellence in Nutrition and Feeding Difficulties, Sabará Children’s Hospital, PENSI Institute, São Paulo 012228-200, Brazil; mauro.fisberg@gmail.com; 2Department of Family Relations and Applied Nutrition, University of Guelph, Guelph, ON N1G2W1, Canada; jhaines@uoguelph.ca; 3Department of Nutrition, School of Public Health, University of São Paulo, São Paulo 01246-904, Brazil; regina.fisberg@gmail.com (R.M.F.); alinevm3@hotmail.com (A.V.d.M.); chscris@yahoo.com.br (C.H.S.); 4Universidad de Santiago de Chile (USACH), Escuela de Ciências de la Actividad Física, el Deporte y la Salud, Santiago 750-0618, Chile; gersonferrari08@yahoo.com.br; 5Center of Molecular Nutrition and Chronic Diseases, Department of Nutrition, Diabetes, and Metabolism, School of Medicine, Pontifical Catholic University of Chile, Santiago 833-0024, Chile; arigotti@med.puc.cl; 6Department of Biochemistry, School of Medicine, University of Costa Rica, San José 11501-2060, Costa Rica; georginagomezcr@gmail.com; 7Faculty of Medicine, Pontifical Catholic University of Argentina, Buenas Aires C1107AAZ, Argentina; ikovalskys@gmail.com; 8Commiteee of Nutrition and Wellbeing, International Life Science Institute (ILSI-Argentina), Buenos Aires C1059ABF, Argentina; 9Department of Nutrition and Biochemistry, Pontifical Univeristy of Javeriana, Bogota 1101111, Colombia; ycortes@javeriana.edu.co; 10Center for Development Studies, Central University of Venezuela/Bengoa Foundation, Caracas 1010, Venezuela; manyma@gmail.com; 11College of Health Science, University of San Francisco Quito, Quito 17-1200-841, Ecuador; myepez@usfq.edu.ec; 12Nutrition Research Institute, La Molina, Lima 15026, Peru; rpareja@iin.sld.pe

**Keywords:** Latin Americans, food sources, shortfall nutrients, diet intake, ELANS, cross-sectional

## Abstract

Increased consumption of energy-dense, nutrient-poor foods can lead to inadequate intakes of shortfall nutrients, including vitamin A, D, C, and E, dietary folate, calcium, iron, magnesium, potassium, and fiber. The objective was to examine the prevalence of inadequate intake of shortfall nutrients and identify food sources of shortfall nutrients in eight Latin American countries. Data from ELANS, a multi-country, population-based study of 9218 adolescents and adults were used. Dietary intake was collected through two 24 h Recalls from participants living in urban areas of Argentina, Brazil, Chile, Colombia, Ecuador, Peru, and Venezuela. Foods and beverages were classified using the adapted version of the NHANES “What We Eat in America” system. Nutrients inadequacy was estimated using the Institute of Medicine recommendations and descriptive statistics were calculated. Prevalence of inadequacy was above 50% for most of the nutrients, which the exception of vitamin C with a prevalence of inadequacy of 39%. Milk, cheese, seafoods, breads, and fruit juices/drinks were among the top 5 sources for each of the 10 shortfall nutrients examined. Many food categories were top contributors to more than one dietary component examined. Understanding the nutrient intake and food sources can help inform dietary guidance and intervention approaches.

## 1. Introduction

Dietary factors are leading contributors to poor health, unhealthy weight gain, and chronic disease risk factors worldwide, including in Latin America [1,2]. The Latin American area includes twenty countries, and over 569 million people; most are low- or middle-income countries. The majority of the Latin American countries are experiencing a nutrition transition marked by undernutrition as well as increasing obesity and other chronic non-communicable diseases [3,4,5]. Although increasing studies on diet intake in recent years [6,7], few of them have examined the diet and food intake among Latin American countries. 

While many Latin American countries have food-based dietary guidelines that promote the intake of nutrient-dense foods, such as fruit and vegetables, eggs, and milk and milk products, [1] many Latin Americans are not meeting their nutrients needs [8,9,10]. Therefore, understanding how Latin Americans are eating and their food sources of various nutrients can provide foundational knowledge to help inform public health policies and behavioral change strategies aimed at improving Latin American nutritional outcomes.

Cross-country analyses based on dietary data from the Latin American Study of Health and Nutrition (ELANS) have previously demonstrated a high intake of total energy intake [11], total sugars, and added sugars [12] among 6648 adolescents and adults (15–65 years old) from eight Latin American countries: Argentina, Brazil, Chile, Costa Rica, Colombia, Ecuador, Peru, and Venezuela. The high consumption of energy-dense, nutrient poor foods, suggest an inadequate intake of fruit, vegetables, grains, and milk [13]. This may result in a shortfall of nutrients, including vitamin A, D, E, and C, folate, calcium, magnesium, potassium, and fiber. For female adolescents and women of childbearing age, iron also is a shortfall nutrient. These nutrients have been identified as nutrients of public health concern, because their underconsumption has been linked to adverse health outcomes [14]. 

Previous analyses of Latin American countries have identified only major groups of energy-intake [11] and nutrients-to-limit sources (i.e., added sugars, saturated fats, and sodium) [13] in adolescents and adult populations. Therefore, there is a lack of studies that identify shortfall nutrients food sources in representative studies in Latin America. The availability of multi-center, representative data from the Latin American Study of Health and Nutrition (ELANS) provides a unique opportunity to investigate the top food and beverages sources of Latin Americans’ intake of shortfall nutrients of public health concern.The objectives of this study were to examine the prevalence of inadequate intake of shortfall nutrients and identify the food sources of shortfall nutrients in eight Latin American countries: Argentina, Brazil, Chile, Colombia, Costa Rica, Ecuador, Peru, and Venezuela. 

## 2. Materials and Methods

Analyses were conducted using the Latin American Study of Health and Nutrition/Estudio Latino Americano de Nutrición y Salud (ELANS) [15] at the University of São Paulo. The ELANS protocol was approved by the Western Institutional Review Board (n° 20140605) and registered in Clinicaltrials.gov (n° NCT02226627). Local research institutes ethics institutional review board from each country also approved the study. All participants provided written informed/assent consent prior to their participation in the study.

### 2.1. ELANS Overview

ELANS is a multi-country, cross-sectional study that uses a multistage stratified area of probability sample of non-institutionalized individuals that provides representative estimates of the urban Latin American population [15]. ELANS aimed to describe the weight status and lifestyle behaviors (diet intake, physical activity, sedentary time and sleep) representing 80–90% of the urban population of Argentina, Brazil, Chile, Colombia, Costa Rica, Ecuador, Peru, and Venezuela. Data collection was conducted from September 2014 to July 2015. Methods for this study are consistent with the protocols of the ELANS study as described in previous publication [15], along with detailed references on the methods used in this study. 

### 2.2. Sample Study

Data from adolescents and adults from 15 to 65 years old participating in ELANS were used, resulting in a sample of 9218 (52% female). This was a random complex multistage sample, stratified by geographic region, sex, age, and socio-economic status (SES), with random selection of primary and secondary sampling units. The households were selected within each secondary sampling unit, via systematic randomization. The choice of the participant within a household was conducted with 50% of the sample next birthday [16], 50% last birthday [17] methods, controlling for sex, age, and SES. The representative sample size was established with a confidence level of 95% and a maximum error of 3.49%. Sample weighting was applied for each country. 

### 2.3. Dietary Intake Assessment 

Dietary intakes were obtained with two 24 h recall (24hR) at the participant home on non-consecutive days, with an interval of up to 8 days between them. The interview was administrated using an automated multiple-pass method [18]. Trained dietitians collected the recall data in Spanish or Portuguese, as appropriate. A detailed description of the dietary interview method is provided elsewhere [11,19]. In summary, foods and beverages reported in the 24hR were converted to energy, macro and micronutrients values using the Nutrition Data System for Research (NDS-R, version 2013) [20]. Because the NDS-R uses the US Department of Agriculture (USDA) food composition database, a standardized procedure matching local foods to USDA foods was conducted by registered dietitians in each country to minimize errors and verify quantities of key nutrients [21]. Food sources of shortfall nutrients vitamins A, D, E, and C, calcium, magnesium, fiber, folate, iron, and potassium were analyzed. In addition, iron was analyzed, since it is considered a shortfall nutrient for female adolescents and childbearing woman [14]. 

The purpose of collecting two 24hRs was to estimate usual dietary intake and evaluate intra-individual variability in nutrient intakes. To estimate nutrient intake, the Multiple Source Method (MSM) technique was applied (https://msm.dife.de) (accessed on 20 April 2021) [22]. The estimation of usual intakes was conducted individually for each country to minimize errors derived from the method. 

### 2.4. Categorization of Food Sources 

The *What We Eat in America* (WWEIA) food category classification system based on the NHANES database [23] was adapted to the Latin American context. Details on the adaptation of this classification system is published elsewhere [13]. Briefly, to adapt to the Latin American context, all the foods consumed were verified by researchers/dietitians in each of the eight countries and these were added to each WWEIA food groups, and when necessary, additional food groups were created to report local foods consumed in Latin America (most of them were Mixed Dishes—Latin American, beans-based items). To retain its international comparability, the foods were kept in each of the groups even though they were not necessarily most frequently consumed or are not available in the Latin American countries (e.g., egg/breakfast sandwich, diet sports/energy drinks, and different types of milk according to fat contain—usually in these countries there are only three types of milk: whole (3%), reduced-fat (0.6–2.9%), and non-fat milk (≤0.5%)). Fruits widely consumed in the Latin American countries were kept in the category “other fruits/fruit salads”. The food group, baby foods/beverage, was excluded because they were not commonly consumed in this target population. From these fourteen main groups; forty-two subgroups (e.g., “bread, rolls, tortillas”; “100% juices”; and “fruits”); and one-hundred and nine categories (e.g., “yeast breads”, “citrus juice”, and “peaches and nectarines”) were included to determine the rank order of contributors to total energy, total grams, and nutrient-to-limits (added sugars, saturated fat, and sodium) of foods/beverages consumed. 

### 2.5. Sociodemographic and Weight Status Variables 

Participants were grouped into 3 age categories (15–19 years; 20–59 years; ≥60 years), with stratification by sex (male and female). Socio-economic status was evaluated by questionnaire using a country-dependent format and based on the legislative requirements or established local standard layouts. This was evaluated creating a categorical (low, middle, and high-income) variable on the basis of the low-income measure, which compares the equivalized per-person income of each country/household with established thresholds for Latin Americans, drawn from national indexes used in each country [15]. 

Weight status was calculated based on participants height and weight measured in duplicate according to the procedures proposed by the WHO [24]. Body Mass Index (BMI) was calculated as body weight (kg) divided by height in meters squared (m^2^). The WHO reference was used to determine weight status in adults [24]: underweight (BMI < 18.5 kg/m^2^), normal weight (BMI = 18.5–24.99 kg/m^2^), overweight (BMI = 25.0–29.99 kg/m^2^), and obesity (BMI ≥ 30 kg/m^2^). Adolescents were classified according to the WHO z-scores for age and sex of each participant [25], considering < −2sd for thinness; > −2 to < +1sd for normal weight, > +1 to < +2sd for overweight, and > +2sd obesity. 

### 2.6. Statistical Analyses

Analyses were conducted using SAS Studio 3.8 (SAS Institute Inc., Cary, NC, USA, 2012–2018). Vitamin A, C, D, and E, folate, calcium, magnesium, fiber, potassium, and iron inadequacy prevalence (%) were estimated according to the Institute of Medicine/Dietary References Intake (IOM/DRI) [26]. Descriptive statistics (means and percentages, with their standard error) for food sources on a population level using SAS PROC RATIO were determined for vitamin A, C, D, and E, folate, calcium, magnesium, fiber, potassium, and iron consumed reported on the day of the 24 h dietary recall. Mean per capita shortfall nutrients consumed from each food group were expressed as percentage of the total to allow relativity across sex and age groups. 

## 3. Results

### 3.1. Demographics and Lifestyle Characteristics 

The sample from the ELANS included 9218 (52% female) respondents aged 15 to 65 years old. By country, the top three countries evaluated in the study were: 21.70% from Brazil, 13.73% in Argentina, and 13.24% in Colombia. BMI (mean ± SE) was 26.91 ± 0.01 kg/m^2^, with 37.14% defined as normal weight, 34.39% as overweight, and 25.14% as obese. A large proportion of the participants were mixed-raced (48.44%), with 52.03% reporting a low-income SES, and 47.66% being married/living with a partner. The average energy intake was 1999.21 ± 5.23 kilocalories (Table 1).

### 3.2. Intake of Shortfall Nutrients 

The prevalence of inadequacy for all the shortfall nutrients for each of the eight Latin American countries and overall sample is shown in Appendix A. On average, the participants were below the recommendations for the most of the shortfall nutrients of public health concern. The prevalence of inadequacy was above 50% for most of the nutrients, which the exception of vitamin C with a prevalence of inadequacy of 39%. The percent of participants meeting the recommendation for potassium was below 7.2% for males and 6.9% for females aged 51–65 years. For iron, the percent of participants meeting the recommendation was 4% for males aged 15–18 years and 20.7% for females aged 19–50 years.

### 3.3. Food Sources of Shortfall Nutrients of Public Health Concern

Table 2 shows the food sources as percentages of all shortfall nutrients of public health concern consumed from the adapted “What We Eat in America (WWEIA)” food categories: vitamin A, D, E, and C; folate, calcium, magnesium potassium, fiber, and iron for the ELANS population. Food sources per country are reported in the Appendix A. 

### 3.4. Food Sources of Vitamin A Intake 

The five highest ranked food categories for the overall ELANS sample contributed 32% of total vitamin A intake. The percentage contribution of each of these food categories was as follow: 12.6% for liver and organ meats, 5.3% milk, whole, 5.0% margarine, 4.9% carrots, and 4.2% other fruit juices. Brazil has the highest contribution with 61.5% as followed by liver and organ meats (31.9%), other starchy vegetables (8.5%), lettuce and lettuce salads (7.5%), margarine (7.3%), and carrots (6.1%). Chile, on the other hand, has the lowest contribution (42.2%) with 16.9% for lettuce and lettuce salads; 6.6% dips, gravies, other sauces; 6.5% eggs and omelets; 6.4% carrots, and 5.7% cheese.

### 3.5. Food Sources of Vitamin D Intake 

The five highest food categories contributed 58.6% of vitamin D consumed in the overall ELANS sample. The food sources were: 26.3% for milk, whole; 10.7% fish; 8.9% eggs and omelets; 6.5% beef, excludes ground; and 6.2% seafood mixed dishes (e.g., stew fish with palm oil and vegetables “*Moqueca de Peixe*”; and breaded fish). Peru contributed with 78.5% being the country with highest consumption as follows the categories fish (45.4%); seafood mixed dishes (12.5%—“*Ceviche*”—raw fish cured on lemon juices); eggs and omelets (9.7%); milk, whole (8.3%); and yogurt, regular (2.6%). Argentina presented the lowest contribution with 47.0%; and categories were 14.5% of milk, whole; 9.1% eggs and omelets; 8.8% turnovers and other grain-based items (e.g., tuna pie); 7.5% milk, reduced fat; and 7.2% fish. 

### 3.6. Food Sources of Vitamin E Intake 

The five highest food categories for the percentage of vitamin E consumed contributed 57.2% of the total ELANS sample and included the following items 44.5% tea; 3.8% margarine; 3.6% meat mixed dishes (e.g., breaded meat); 2.9% beans, peas, legumes; and 2.3% rice mixed dishes (e.g., fried rice without meat “*guiso de arroz*”). Argentina contributed with the highest percentage (82.5%) as the food sources; 62.8% tea, 6.1% salad dressing and vegetable oils, 5.3% meat mixed dishes; 4.9% turnovers and other grain-based items (e.g., meat pies “*empanada de carne*”); and 3.2% poultry mixed dishes (e.g., breaded chicken). Colombia percentage of top 5 food sources was 31.1%, being the country with the lowest contribution. Eggs and omelets contributed with 7.8%, meat mixed dishes (e.g., meat stew with taro, yams, potato, and pumpkin—“*Sancocho*”) 6.9%, fruit drinks 5.8%, banana 5.4%, and potato chips 5.2%. 

### 3.7. Food Sources of Vitamin C Intake 

The top five categories of ELANS sample contributed to 56.9% and food sources were 17.4% for other fruit juices; 13.3% other fruit and fruit salads; 11.8% citrus juice; 10.8% fruit drinks; and 3.7% banana. Brazil contributed the most with 84.9% and food and beverages sources were other fruit juices (48.7%), other fruit and fruit salads (15.9%), citrus juice (14.2%), tomatoes (3.2%), and fruit drinks (2.9%). Costa Rica has the lowest contribution with 53.1% represented by other fruit and fruit salads (23.8%), citrus juice (13.3%), other vegetables and combinations (6.1%), tomatoes (5.0%), and other red and orange vegetables (4.9%).

### 3.8. Food Sources of Dietary Folate Intake 

The ELANS food sources contributed to 48.7% of folate consume and included 14.7% of yeast bread; 11.1% pancakes, waffles, and French Toast; 9.3% rice; 8.7% beans, peas, and legumes; and 4.9% pasta, noodles, and cooked grains. Chile contributed the most with 61.6% of following food sources: 31.7% yeast breads; 14.1% fish; 6.8% pasta, noodles, and cooked grains; 4.9% other fruit and fruit salads; and 4.0% lettuce and lettuce salads. Costa Rica contributed the least with 56.1%, represented by 21.9% beans, peas, and legumes; 11.2% rice; 10.9% yeast breads; 9.1% other Mexican Mixed Dishes; and 3.1% other fruit and fruit salads.

### 3.9. Food Sources of Calcium Intake 

Overall, 43.7% of contributed to calcium intake, represented by cheese (14.0%), milk, whole (13.2%), pizza (6.5%), yeast breads (5.3%), and pancakes, waffles, and French Toast (4.5%). The lowest contribution was 41.9% in Peru represented by 11.1% of cheese, 10.6% of milk, whole, 8.9% yeast breads, 5.9% sugars and honey, and 5.4% rice. The highest contribution was 63.3% in Venezuela being 34.7% for cheese, 11.2% pancakes, waffles, French Toast, 9.1% turnovers and other grain-based items, 4.1% milk, whole, and 4.0% yeast breads. 

### 3.10. Food Sources of Magnesium Intake 

The ELANS sample top five food sources of magnesium contributed to 29.2% of intake, characterized as 9.2% of beans, peas, legumes; 6.2% pancakes, waffles, and French toast; 5.2% rice; 4.5% yeast breads; and 4.1% coffee. Costa Rica contributed to 44.6% of magnesium food sources intake, including beans, peas, and legumes (17.6%), rice (10.2%), other Mexican mixed-dishes (8.1%), coffee (4.5%), and yeast breads (4.2%). The lowest country that contributes to magnesium food sources were Colombia with 29.8% and included 7.1% for bananas, 6.6% rice, 5.9% beans, peas, legumes, 5.3% fruit drinks, and 4.9% coffee. 

### 3.11. Food Source of Potassium Intake 

The top five food sources of potassium contributed to 28.5% in the ELANS sample, including beans, peas, and legumes (9.0%), banana (5.8%), beef, excludes ground (5.7%), milk, whole (4.8%), and soups (3.14%). Brazil contributed the most with 37.5%, represented by 13.2% beans, peas, and legumes; 9.1% beef, excludes ground; 6.8% milk, whole; 4.5% chicken, whole pieces; and 3.9% other fruit and fruit salads. Argentine contributed the least with 28.6%, with top five source including, 6.2% for meat mixed dishes (e.g., breaded meat), 5.9% turnovers and other grain-based (e.g., spinach and chard pie), 5.7% tea, 5.5% beef, excludes ground, and 5.4% yeast breads. 

### 3.12. Food Sources of Fiber Intake 

The top five food sources of fiber contributing to 49.4% for the ELANS total sample and included tea (19.6%), beans, peas, and legumes (13.9%), pancakes, waffles, and French toast (5.5%), yeast breads (5.4%), and soups (5.1%). Argentina contributed the most with 70.4% of sources, representing 53.9% of tea (e.g., “Herb tea”—*Yearba Mate*), 8.0% yeast breads, 3.3% other fruit and fruit salads, 2.6% turnover and other grain-based (e.g., spinach and chard pie), and 2.6% pizza. Peru contributed the least with 40.0% and sources included 8.8% yeast breads; 8.4% soups (e.g., legume soup with vegetable—“*Sopa de morón con verduras*”); 8.4% other fruit and fruit salads; 8.1% beans, peas, and legumes; and 6.5% white potatoes, baked or boiled. 

### 3.13. Food Sources of Iron Intake

The top five sources of iron intake contribute to 41.6% of the ELANS sample, including 9.4% yeast breads; 9.3% rice; 9.2% beans, peas, and legumes; 8.9% pancakes, waffles, and French Toast; and 4.7% beef, excludes ground. Brazil contribution of five food sources were 54.3% representing 16.2% yeast breads; 13.7% beans, peas, and legumes; 11.4% rice; 9.4% beef, excludes ground; and 3.6% chicken, whole pieces. On the order hand, Colombia contributed to 38.0% as following food sources: 12.1% rice; 9.2% yeast breads; 7.6% beans, peas, and legumes; 4.8% meat mixed dishes (e.g., meat stew with taro, yams, potato, and pumpkin—“*Sancocho*”); and 4.4% beef, excludes ground. 

## 4. Discussion

These analyses were conducted to help understand the major sources of shortfall nutrients of public health concern among Latin American individuals. Overall, the prevalence of inadequate intake was over 50% for most shortfall nutrients, highlighting the importance of efforts to support healthy eating within Latin America. Moreover, our results show major food sources overlap among dietary components. Milk, cheese, fish/seafoods, breads, and fruit juices/drinks were among the top 5 sources for each of the 10 shortfall nutrients examined in the ELANS sample overall and for each of the eight countries. 

Although considerable attention has been reported on sugar-sweetened beverages (SSB) and their role in excess weight gain [27,28], some of the items of these broaden beverage category can provide important sources of shortfall nutrients, i.e., vitamin E and C, and magnesium [6]. Thus, guidance on intake inclusion of fruit juices and drinks in this definition is important for the improvement of these shortfall nutrients in the population diet [29] through the development of successful strategies integrating the tripartite: industry, government, and academia. The totality of nutrients provided by foods or beverages have to be considered and balanced with food sources of shortfall nutrients when making dietary recommendations to build a healthy diet [6]. For example, dietary patterns that encourage enriched or fortified food groups may help shift population consumption in Latin America populations toward recommended intake levels for several shortfall nutrients identified by the 2015 concensus on dietary guidelines [30]. However, fortified/enriched foods can present both a shortfall and overconsumed nutrients of public health concern [6,31]. Thus, overconsumption for these food sources can lead to excessive energy intake, and might lead to unhealthy weight gain [5]. 

While the Latin American sample are not meeting the recommendations for the majority of the shortfall nutrients, many of their top food sources retain important nutrients of public health concern. The ELANS sample are consuming as major sources of home-based mixed-dishes (e.g., preparations including beans, vegetables, and soups), fruits and vegetables, and/or processed items (e.g., fruit drinks, milk, and cheese). There is a dire need to strengthen the design, implementation, and evaluation of strategies to increase access to and promote healthful items in the Latin America countries. For example, updates and revisions of current dietary guidelines for Latin American countries, may be needed. This includes the design and promotion of messages based on day-by-day eating patterns of different groups of individuals. Adopting strategies that focus on tradition and local foods, as well as promoting healthful food option within industrialized items may help improve dietary intake [32]. An example for all Latin American countries could be to promote an increase on servings of plant-based proteins food items, given the variety on the consumption of these through tradition and local food sources. Other dietary guidelines, such as the new Canada Food Guide, includes a focus on plant-based proteins [33]. Increasing the intake of plant-based proteins has been proposed as potential strategy for improving diet quality and reducing risk of cardiometabolic disease [32].

A key strength of these analyses was the use of nationally representative data and the use of two 24 h recalls to estimate the usual dietary intake of the sample. However, some limitations should be noted. First, 24 h dietary recalls have inherent limitations, including errors due to memory, underreporting of energy, and examiner effects [6]. Second, caution should be used in making direct comparisons between food sources rankings in this publication with other published work because of the dependence of rankings on food grouping definitions. 

## 5. Conclusions

Overall, these analyses provide an understanding of major food sources of shortfall nutrients: vitamin A, D, C, and E, dietary folate, calcium, magnesium, potassium, fiber, and iron; in the Latin American population, including categories of foods and beverages that are key contributors to more than one dietary shortfall component. Understanding the nutrient intake and food sources is crucial to informing dietary guidance to policy makers, researchers, practitioners, and, other stakeholders to develop effective public health policies and behavioral change strategies to improve dietary intake. Efforts to continue monitoring the food sources of nutrients by Latin Americans and other HIC and LMIC populations should be an important aspect to increase the current knowledge in this field. 

## Figures and Tables

**Table 1 ijerph-18-04967-t001:** Demographics and lifestyle characteristics of the Latin American population. Latin American Health and Nutrition Study/Estudio Latino Americano de Nutrición y Salud (ELANS).

Characteristics	*n* (%)
Country	
Argentina	1266 (13.73)
Brazil	2000 (21.70)
Chile	879 (9.54)
Colombia	1230 (13.24)
Costa Rica	798 (8.66)
Ecuador	800 (8.68)
Peru	1113 (12.07)
Venezuela	1132 (12.28)
Sex	
Female	4809 (52.17)
Male	4409 (47.83)
Income Status	
High	880 (9.55)
Medium	3542 (38.42)
Low	4696 (52.03)
Marital Status	
Single	3912 (42.44)
Married or living with a partner	4393 (47.66)
Widow	235 (2.55)
Divorced	678 (7.36)
Race/Ethnicity	
Mixed-Race	4240 (48.44)
Caucasian	3216 (36.74)
African-American	995 (11.37)
Native	178 (2.03
Other ^a^	124 (1.42)
Weight Status	
Underweight	306 (3.32)
Normal weight	3420 (37.14)
Overweight	3167 (34.39)
Obese	2315 (25.14)
	M ± SE
Age	35.82 ± 0.15
BMI (kg/m^2^)	26.91 ± 0.01
Total Energy Intake (Kcal/day)	1999.21 ± 5.23

^a^ Other: gypsy, Asian, and other ethnicities. SE: Standard Error.

**Table 2 ijerph-18-04967-t002:** Food sources of shortfall nutrients intake among Latin American adolescents and adults (*n* = 9218). The Latin American Study of Nutrition and Health.

Main Group	Subgroup	Categories	Cons ^1^	Mean ± SE	%
**Vitamin A (µg/day)**					
Protein Foods	Meats	Liver and organ meats	225	230.2 ± 369.5	12.6
Milk and Dairy	Milk	Milk, whole	2382	86.5 ± 1.2	5.3
Fats and Oils	Fats and Oils	Margarine	1672	117.2 ± 2.7	5.0
Vegetables	Vegetables, excluding potatoes	Carrots	371	513.3 ± 23.8	4.9
Beverages	100% juice	Other Fruit Juice	1013	175.1 ± 8.3	4.2
**Vitamin D (mg/day)**					
Milk and Dairy	Milk	Milk, whole	2382	2.42 ± 0.03	26.3
Protein Foods	Seafood	Fish	528	4.78 ± 0.32	10.7
Protein Foods	Eggs	Eggs and Omelets	1506	1.36 ± 0.02	8.9
Protein Foods	Meats	Beef, excludes ground	1882	0.76 ± 0.01	6.5
Mixed Dishes	Mixed Dishes—M/P/S	Seafood Mixed Dishes	311	5.86 ± 0.61	6.2
**Vitamin E (mg/day)**					
Beverages	Coffee and Tea	Tea	3897	6.94 ± 0.15	44.5
Fats and Oils	Fats and Oils	Margarine	1672	1.36 ± 0.03	3.8
Mixed Dishes	Mixed Dishes—M/P/S	Meat Mixed Dishes	781	3.31 ± 0.14	3.6
Protein Foods	Plant-Based Proteins	Beans, peas, legumes	3701	0.49 ± 0.01	2.9
Mixed Dishes	Mixed Dishes—Grains	Turnovers and other grain-based items	598	2.28 ± 0.06	2.3
**Vitamin C (mg/day)**					
Beverages	100% Juices	Other Fruit Juice	1013	88.21 ± 2.61	17.4
Fruits	Fruits	Other Fruits and Fruit Salads	1056	62.88 ± 2.18	13.2
Beverages	100% Juices	Citrus Juice	1245	47.53 ± 2.22	11.8
Beverages	Sweetened Beverages	Fruit Drinks	1450	35.03 ± 1.06	10.8
Fruits	Fruits	Banana	1421	12.32 ± 0.28	3.7
**Folate DFE (mg/day)**					
Grains	Bread, Rolls, Tortillas	Yeast Breads	3465	128.00 ± 1.63	14.7
Grains	Quick Breads and Bread Products	Pancakes, waffles, French Toast	2134	153.81 ± 1.81	11.1
Grains	Cooked grains	Rice	4680	59.82 ± 0.61	9.3
Protein Foods	Plant-based Protein Foods	Beans, peas, legumes	3701	69.87 ± 1.36	8.7
Grains	Cooked Grains	Pasta, noodles, cooked grains	594	82.11 ± 1.36	8.7
**Calcium (mg/day)**					
Milk and Dairy	Cheese	Cheese	1932	290.07 ± 3.98	14.0
Milk and Dairy	Milk	Milk, whole	2382	223.14 ± 3.00	13.2
Mixed Dishes	Mixed Dishes—Pizza	Pizza	372	706.77 ± 34.45	6.5
Grains	Bread, rolls, tortillas	Yeast Breads	3465	62.84 ± 1.10	5.3
Grains	Quick breads and bread products	Pancakes, Waffles, French Toast	2134	85.90 ± 3.13	4.5
**Magnesium (mg/day)**					
Protein Foods	Plant-based Protein Foods	Beans, peas, legumes	3701	37.64 ± 0.55	9.2
Grains	Quick Breads and Bread Products	Pancakes, waffles, French Toast	2134	43.88 ± 0.50	6.2
Grains	Cooked grains	Rice	4680	17.02 ± 0.20	5.2
Grains	Bread, Rolls and Tortillas	Yeast breads	3465	13.22 ± 0.23	4.5
Beverages	Coffee	Coffee	4135	14.78 ± 0.29	4.1
**Iron (mg/day)**					
Grains	Bread, Rolls, Tortillas	Yeast Breads	3465	2.28 ± 0.02	9.4
Grains	Cooked grains	Rice	4680	1.68 ± 0.02	9.3
Protein Foods	Plant-based Protein Foods	Beans, Peas, Legumes	3701	2.09 ± 0.03	9.2
Grains	Quick bread and bread products	Pancakes, Waffles, French Toast	2134	3.48 ± 0.04	8.9
Protein Foods	Meats	Beef, excludes ground	1882	2.09 ± 0.03	4.7
**Potassium (mg/day)**					
Protein Foods	Plant-based protein foods	Beans, peas, legumes	3701	320.4 ± 4.01	9.0
Fruits	Fruits	Banana	1421	522.81 ± 11.82	5.8
Protein Foods	Meats	Beef, excludes grounds	1882	398.68 ± 5.51	5.7
Milk and Dairy	Milk	Milk, whole	2382	262.06 ± 3.54	4.8
Mixed Dishes	Mixed Dishes—Soups	Soups	1459	349.18 ± 8.15	3.1
**Fiber (mg/day)**					
Beverages	Coffee and Tea	Tea	3897	5.81 ± 0.13	19.6
Protein foods	Plant-based protein foods	Beans, peas, legumes	3701	4.35 ± 0.07	13.9
Grains	Quick breads, and bread products	Pancakes, waffles, French Toast	2134	2.95 ± 0.03	5.5
Grains	Bread, rolls, tortillas	Yeast breads	3465	1.81 ± 0.02	5.4
Mixed Dishes	Mixed Dishes—Soups	Soups	1459	4.96 ± 0.15	5.1

Note: Cons: consumption; DFE: Dietary Folate Equivalent; M/P/S: Meat/Poultry/Seafoods; SE: Standard Error. ^1^ Cons: number of times food have been consumed by the entire population.

## Data Availability

No new data were created or analyzed in this study. Data sharing is not applicable to this article.

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
