# Peer review of "Food Sources of Shortfall Nutrients among Latin Americans: Results from the Latin American Study of Health and Nutrition (ELANS)"

_ijerph, 2021, doi:10.3390/ijerph18094967_

Round 1
Reviewer 1 Report
Dear Authors,
The manuscript (ijerph-1187828) presented for review is very interesting and I recommend the article for publication. This manuscript is well written. I have only one suggestion connected to the title of this manuscript.
In my opinion, the title should be rather: Food sources and shortfall of vitamin, minerals, and fiber in the diet. Results from the Latin American Study of Health and Nutrition (ELANS). Due to the fact that study results are connected only to these nutrients.
The second comment is connected with section Discussion.
In section Discussion, the authors didn't discuss their results with other previous research from Latin America. Although this group of respondents is very big (n=9218), it is specific because 59.5% of the population was overweight and obese. It is interesting to know did previous studies indicate other deficiencies or maybe the Latin American diet change for example during two, three last decades?
Despite my comments, I am pleased to recommend this manuscript for publication.
Reviewer
Author Response
Reviewer 1
In my opinion, the title should be rather: Food sources and shortfall of vitamin, minerals, and fiber in the diet. Results from the Latin American Study of Health and Nutrition (ELANS). Due to the fact that study results are connected only to these nutrients.
Response: Thank you for your suggestion. Since we were using an adapted classification system that classifies foods into two major groups: overconsumed and shortfall nutrients of public health concern. The major name of the groups are nutrients of public health concern, and then from this, we have overconsumed (which are added sugars, saturated fats, and, sodium), and the other is shortfall nutrients (which are the vitamin A, C, D, E, folate, calcium, magnesium, fiber, and, potassium, and iron for adolescent girls and childbearing women). Thus, saying “Food sources and shortfall of (…)” seems incorrect according to language use. See below the reference documentation where this classification was referred:
Committee DGA (2020) Scientific Report of the 2020 Dietary Guidelines Advisory Committee. Advisory Report to the Secretary of Agriculture and Secretary of Health and Human Services. U.S. Department of Agriculture, Agricultural Research Service, Washington, DC.
As well as other publications that used this classification (and the given title):
Leme AC, Baranowski T, Thompson D, Philippi S, O'Neil CE, Fulgoni VL, 3rd, Nicklas TA (2020) Food Sources of Shortfall Nutrients Among US Adolescents: National Health and Nutrition Examination Survey (NHANES) 2011-2014. Fam Community Health 43 (1):59-73. doi:10.1097/FCH.0000000000000243
O'Neil CE, Nicklas TA, Fulgoni VL, 3rd (2018) Food Sources of Energy and Nutrients of Public Health Concern and Nutrients to Limit with a Focus on Milk and other Dairy Foods in Children 2 to 18 Years of Age: National Health and Nutrition Examination Survey, 2011(-)2014. Nutrients 10 (8). doi:10.3390/nu10081050
Papanikolaou Y, Fulgoni VL (2018) Grains Contribute Shortfall Nutrients and Nutrient Density to Older US Adults: Data from the National Health and Nutrition Examination Survey, 2011(-)2014. Nutrients 10 (5). doi:10.3390/nu10050534
The second comment is connected with section Discussion.
In section Discussion, the authors didn't discuss their results with other previous research from Latin America. Although this group of respondents is very big (n=9218), it is specific because 59.5% of the population was overweight and obese. It is interesting to know did previous studies indicate other deficiencies or maybe the Latin American diet change for example during two, three last decades?
Response: Thank you for your suggestion. However, for this study, the aim was to assess the shortfall nutrients of public concern (i.e., vitamin A, C, D, E, calcium, folate, magnesium, potassium, and iron). The prevalence of overweight/obese was not the purpose of this study, as well identification of associations between weight status and shortfall of nutrients. We thought that we should not mix things on this behalf. However, we added on the discussion, which we agree with you, that overweight/obesity might be a concern, and this might lead to nutrients deficiencies. Also, it is hard to find some publications, specifically from Latin American countries, that reported the prevalence of inadequacies and overweight/obesity due to lack of unstandardized measurement tools. Hence, we added similar studies (although from HIC) to reinforce the need for more standardized measurements in LA in order to promote the international generalizability of the studies.
Reviewer 2 Report
Thank you for the opportunity to review this manuscript. This is an interesting study which provides a look at the prevalence of inadequate nutrients among 8 Latin American countries. It is well-written, but there are some areas of the manuscript that could be strengthened and clarified. Below are some comments to consider:
Abstract: You only list 7 countries here, what is the 8th? You mentioned Costa Rica in the introduction. Also, you might briefly define what prevalence of inadequacy means (or for example, what is a “good” level of inadequacy)
Line 53: Do you mean “includes” here?
Line 56: What is increasing? Studies of diet and food intake? Please specify and cite if needed.
Lines 69-70: You mention :”might suggest poor intake” and “might indicate a shortfall”, but is there evidence that either is the case? For example, is there evidence that high energy foods and sugar intake is displacing fruits/vegetables in Latin America? I would recommend finding stronger evidence and citing that here to create a more powerful introduction
Line 73: DO you mean “women”? Also, would it be better to say “women of childbearing age”?
Lines 65-75: I think more could be added in this section to demonstrate a shortage of the nutrients in Latinx countries. Is there additional data that can be cited from the ELANS study? This section sets the tone for the gap this study is filling and currently, it seems weak. You could mention here what is a healthy or good level of inadequacy countries should strive for, and any evidence that inadequacy is high in Latin America
Line 84: Is this a secondary analysis? If so, just state that.
Line 87: You might clarify what each country’s ethics committees represented here. Were they the author’s institutions? Were they governmental organizations? Or higher education institutions? Clarity can help if someone wanted to replicate this study of 8 countries.
Line 92: What was the remaining 10-20% of the population not represented in the ELANS study? You might clarify who is being represented here.
Line 101: Again, just state if this is a secondary analysis.
Line 15): You might add how you calculate BMI and what cutpoints were used in this section.
Line 151: Is there a justification for these age categories? Just curious. If so, please cite what method was used to guide the division of age
Lines 170-177: You could remove some info from either this text or in Table 1. For example you could delete the percent by each country in the text (or just list the top 2-3 countries) and refer the reader to the table for full results. Just a thought.
Table 2: What is the heading “Cons”? Can you clarify?
Conclusions: What do researchers, academia, dietitians, governments, etc. do with the findings of this study? Will the results inform efforts in any one of the 8 countries? Are there current programs that will use the results? Your statement in lines 337-338 is very general and not as impactful as more specific recommendations for either Latin America or any of the 8 countries. I feel this section could be strengthened.
Author Response
Abstract: You only list 7 countries here, what is the 8th? You mentioned Costa Rica in the introduction. Also, you might briefly define what prevalence of inadequacy means (or for example, what is a “good” level of inadequacy).
Response: Thank you for your comment. Our mistake, we forgot to mention Costa Rica in the abstract. In regards to the “prevalence of inadequacy,” we provided how we define it.
Line 53: Do you mean “includes” here?
Response: Thank you for your edits. We provided the changes.
Line 56: What is increasing? Studies of diet and food intake? Please specify and cite if needed.
Response: We appreciate your comment. Yes, you were correct it was a typo error. We added the missing information and references.
Lines 69-70: You mention :”might suggest poor intake” and “might indicate a shortfall”, but is there evidence that either is the case? For example, is there evidence that high energy foods and sugar intake is displacing fruits/vegetables in Latin America? I would recommend finding stronger evidence and citing that here to create a more powerful introduction.
Response: Thank you. We provided a citation, which includes a recently published article from our group.
Line 73: DO you mean “women”? Also, would it be better to say “women of childbearing age”?
Response: Yes, we mean women of childbearing age. We provided the changes.
Lines 65-75: I think more could be added in this section to demonstrate a shortage of the nutrients in Latinx countries. Is there additional data that can be cited from the ELANS study? This section sets the tone for the gap this study is filling and currently, it seems weak. You could mention here what is a healthy or good level of inadequacy countries should strive for, and any evidence that inadequacy is high in Latin America.
Response: We agree with you. We added more to this section to demonstrate the gap of the nutrients in Latin American countries, as well references to support this.
Line 84: Is this a secondary analysis? If so, just state that.
Response: We appreciate your comment. But this was not a secondary analysis, hence we opt not to add this to the methods section.
Line 87: You might clarify what each country’s ethics committees represented here. Were they the author’s institutions? Were they governmental organizations? Or higher education institutions? Clarity can help if someone wanted to replicate this study of 8 countries.
Response: We appreciate your comment. We provide the changes.
Line 92: What was the remaining 10-20% of the population not represented in the ELANS study? You might clarify who is being represented here.
Response: Thank you for your comment. We provide more information on this. It represents the urban population from the LA countries.
Line 101: Again, just state if this is a secondary analysis.
Response: We provide our response below from your inquiry on “line 84”.
Line 15): You might add how you calculate BMI and what cutpoints were used in this section.
Response: We agree with you. We provide how we calculate BMI and cut-points used.
Line 151: Is there a justification for these age categories? Just curious. If so, please cite what method was used to guide the division of age.
Response: We appreciate your comment. These age categories were based on previous ELANS study hence, we opt to maintain this for consistency over the other published studies. We added the citation.
Lines 170-177: You could remove some info from either this text or in Table 1. For example you could delete the percent by each country in the text (or just list the top 2-3 countries) and refer the reader to the table for full results. Just a thought.
Response: We agree with you. We delete the % by each country in the text and add the reader to refer to table1.
Table 2: What is the heading “Cons”? Can you clarify?
Response: We agree with you. It was missing a footnote to clarify “cons”. We added on table 2.
Conclusions: What do researchers, academia, dietitians, governments, etc. do with the findings of this study? Will the results inform efforts in any one of the 8 countries? Are there current programs that will use the results? Your statement in lines 337-338 is very general and not as impactful as more specific recommendations for either Latin America or any of the 8 countries. I feel this section could be strengthened.
Response: We agree with you. It was missing the link between researchers/government/practitioners on their development for future efforts to improve diet. We made it clear to the section.